# Can anthropomorphic design in beverage packaging enhance impulse buying intention? Amazing visual and verbal cues!

**Yuyang Tian**[1], **Siti Farhana Zakaria**[1]*, **Yinghui Du**[2], **Ye Qiu**[3], **Zenghui Tian**[2]

1 College of Creative Arts, Universiti Teknologi MARA (UiTM), Shah Alam, Selangor, Malaysia, 2 Faculty of Art, Sustainability and Creative Industry, Sultan Idris Education University, Tanjung Malim, Malaysia, 3 College of Arts and Media, Nanning University of Science and Technology, Nanning, China

* farha981@uitm.edu.my

**Data availability statement:** The dataset of this study has been uploaded to the public database Mendeley Data at DOI: 10.17632/tdvycymws7.1 (https://data.mendeley.com/datasets/tdvycymws7/1).

## Abstract

The anthropomorphic design of beverage packaging has gained significant attention as an innovative marketing strategy. This study systematically examines how anthropomorphic visual and verbal cues in beverage packaging influence consumers' impulse buying intentions. Grounded in the Stimulus-Organism-Response model, Social Presence Theory, and Empathy Theory, the research employs a consumer survey to validate the mediating roles of positive emotions, social presence, and empathy in the relationship between anthropomorphic design and impulse buying intentions. A comprehensive causal model was developed to explore these relationships. While modern packaging techniques and visual elements enhance the appearance of beverage packaging, they did not significantly trigger empathetic responses. This suggests that emotional language and personalized expressions are more effective in fostering emotional resonance. The study also highlights the significant mediating roles of positive emotions and social presence in the relationship between anthropomorphic design and impulse buying intentions, emphasizing the importance of emotional and social factors in consumer decision-making.This research validates the theoretical model and offers practical insights for companies. It suggests that businesses should focus on consumers' emotional needs and psychological states in packaging design and marketing strategies. By enhancing emotional language, optimizing visual anthropomorphic elements, and incorporating multi-sensory designs, companies can effectively improve consumers' positive emotions and social presence, thereby increasing impulse buying intentions. For consumers with high levels of loneliness, strengthening anthropomorphic design elements can meet their social and emotional needs, enhancing brand loyalty and market competitiveness.

## 1 Introduction

In recent years, anthropomorphic design has emerged as a prominent marketing strategy, increasingly utilized in the modern consumer market, particularly in product packaging. By attributing human-like characteristics to products, anthropomorphic design in beverage

**Funding:** The author(s) received no specific funding for this work.

packaging can evoke emotional resonance among consumers, enhancing their sense of identification with the product and stimulating purchase intentions. In the Chinese market, attractive packaging is a key design factor ranked among the top purchasing considerations. Compared to other factors such as flavor innovation and pricing, optimizing packaging design appears to be a cost-effective and rapidly impactful marketing approach. For instance, the packaging of Intermarché's orange juice garnered 50 million media impressions within three hours of its launch, and sales surged by 4,600% within three weeks, with global in-store traffic increasing by 25% (DIGITALING). While the design strategy played a significant role, it is important to note that other factors such as brand awareness and promotional activities may have also contributed to this success. Similarly, VDA BANGKOK established TASTE&Co to implement innovative packaging designs, helping small-scale producers in Thailand expand their businesses (Sohu). The global beverage packaging market was valued at 138.72 billion in 2022 and is expected to grow to 204.53 billion by 2030, with a compound annual growth rate of 5.1% from 2023 to 2030 (DATA BRIDGE). This growth trend underscores the importance of innovative packaging design in driving market expansion.

These data suggest that anthropomorphic design in packaging can enhance consumers' emotional identification and purchase intentions [1]. However, the psychological mechanisms underlying consumer behavior in response to anthropomorphic designs remain underexplored. Most existing studies focus on the impact of visual designs on consumers, such as cartoon patterns and anthropomorphic appearances, with limited attention given to the role of verbal cues in anthropomorphic beverage packaging designs. For example, whether anthropomorphic language styles influence consumer behavior through psychological mechanisms such as social presence, positive emotions, and empathy remains insufficiently explored [2]. Therefore, further research is needed to examine the roles of visual and verbal cues in anthropomorphic design and explore how these cues affect consumer behavior through complex psychological mediation mechanisms. Anthropomorphic design is a strategy that evokes emotional resonance in consumers by assigning human traits to products [3]. These traits include anthropomorphic elements in appearance, such as facial expressions and body shapes, as well as emotional and personality expressions conveyed through verbal communication. Despite its widespread application in marketing, the influence of anthropomorphic design on consumer purchase behavior through psychological mechanisms such as affective responses and cognitive evaluation remains underexplored [4]. Grounded in Affective Response Theory, this study examines how anthropomorphic design influences impulse buying intentions through emotional and cognitive pathways [5]. Affective Response Theory posits that consumers' emotional reactions to stimuli, such as packaging design, play a critical role in shaping their behavioral responses.

This study incorporates Affective Response Theory into the framework of anthropomorphic design in beverage packaging, focusing on the roles of visual and verbal cues in evoking positive emotions, enhancing social presence, and driving purchase behavior [6]. Positive emotions play a crucial role in the impact pathway of anthropomorphic design [7]. Feelings of pleasure and satisfaction directly enhance a product's appeal to consumers, making them more inclined to purchase immediately [8]. This study further analyzes how visual and verbal cues in anthropomorphic beverage packaging design influence impulse buying intentions through positive emotions and social presence. The findings aim to provide businesses with refined and emotionally driven packaging design and marketing strategies [9]. By revealing consumer purchasing behavior characteristics in different contexts, the study seeks to advance market design strategies, shifting from mere visual appeal to deeper emotional resonance and social connection. Ultimately, it aims to influence consumer impulse buying intentions through complex psychological mechanisms. Affective Response Theory plays a key role in

understanding anthropomorphic design, revealing how anthropomorphic design influences impulse buying intentions through psychological mechanisms such as social presence, positive emotions, and empathy. According to this theory, affective responses s to stimuli are critical in shaping consumer behavior [10]. Anthropomorphic beverage packaging, which assigns human traits to products, enhances consumers' emotional engagement. Consumers generally exhibit stronger affective responses to objects with human-like traits [11]. Grounded in Affective Response Theory, this study investigates how anthropomorphic design affects impulse buying intentions via emotional and cognitive pathways. Visual cues, such as facial expressions and anthropomorphic images, effectively capture consumer attention, fostering affinity, enhancing social presence, and evoking positive emotions. These emotions, in turn, drive impulse purchases [12]. Verbal cues, including emotional language and humor, convey product personality, eliciting empathy, strengthening emotional connections, and further boosting social presence and purchase intentions [13].

Globally, 54% of leading brands use anthropomorphic elements in packaging, with notable examples being M&M's "spokescandies" and first-person language in product descriptions [14]. While visual cues exert a more immediate and pronounced impact, verbal cues produce deeper cognitive effects, shaping attitudes and beliefs [15]. This research emphasizes the distinct roles of visual and verbal cues, offering theoretical insights for packaging design and marketing strategies, aiding businesses in stimulating impulse buying behavior more effectively. Understanding these mechanisms in greater depth enables businesses to craft more effective marketing strategies and boost sales.

## 2 Theoretical background

### 2.1 Concept of impulse buying intention

Impulse buying intention refers to a consumer's spontaneous and unplanned decision to purchase a product, often triggered by external stimuli [16]. This behavior is characterized by its immediacy, emotional intensity, and lack of deliberate evaluation. Unlike planned purchases, impulse buying is heavily influenced by emotional and environmental factors, making it a central focus in consumer behavior research. In the digital age, the convenience and immediacy of online shopping have further amplified impulse purchase intentions. Positive emotions, such as excitement or joy, play a significant role in enhancing these tendencies [17]. Environmental factors, including product displays, promotional activities, and packaging design, also exert a strong influence. For instance, Zafar et al. (2020) [18] demonstrated that anthropomorphic packaging—which assigns human-like traits to products—can evoke emotional resonance and amplify positive emotions, thereby increasing impulse purchase intentions. The Stimulus-Organism-Response (S-O-R) model provides a robust theoretical framework for understanding the formation of impulse purchase intentions. According to this model, external stimuli (S) influence an individual's internal states (O), which in turn drive behavioral responses (R). In the context of this study, anthropomorphic beverage packaging serves as the external stimulus, utilizing visual and verbal cues to evoke emotional and cognitive responses in consumers. These internal responses, such as heightened positive emotions or a sense of social connection, then lead to increased impulse purchase intentions. The S-O-R model is particularly applicable to this research because it captures the dynamic interplay between environmental stimuli (e.g., anthropomorphic packaging), psychological processes (e.g., emotional resonance and trust), and behavioral outcomes (e.g., impulse buying). For example, Fang et al. (2022) [19] argue that weakened self-control, as explained by self-regulation theory, increases the likelihood of impulse buying. Emotional induction theory further suggests

that positive emotions reduce self-control, making consumers more susceptible to impulsive decisions.

## 2.2 Affective response theory

Affective Response Theory In Affective Response Theory (ART), consumers' emotional reactions to external stimuli play a crucial role in shaping their behavioral responses [20]. Affective responses are immediate and automatic reactions to stimuli, which subsequently influence cognitive evaluations and decision-making processes [21]. ART is widely applied in marketing and consumer behavior research to explain how emotional engagement with products, advertisements, or packaging designs drives consumer actions, such as purchase intentions and brand loyalty [22]. In the context of anthropomorphic design, ART provides a robust framework for understanding how human-like characteristics in product packaging evoke emotional resonance, thereby influencing consumer behavior [23]. By assigning human traits to products, anthropomorphic design creates a sense of social interaction and emotional connection, triggering affective responses that enhance consumers' identification with the product and their willingness to purchase [24]. Anthropomorphic design often elicits feelings of pleasure, joy, and satisfaction. For example, facial expressions or playful language on beverage packaging can create a sense of friendliness and fun, enhancing consumers' emotional engagement with the product [25]. Human-like traits in packaging design can evoke empathetic responses, making consumers feel a deeper emotional connection to the product. This empathy strengthens the emotional bond between the consumer and the product, increasing the likelihood of impulse purchases. Therefore, impulse buying is often driven by emotional factors rather than rational ones.

## 3 Hypotheses

### 3.1 Conceptual model

The conceptual model is shown as Fig 1:

### 3.2 Anthropomorphic design in beverage packaging and impulse buying intention

Anthropomorphic beverage packaging assigns human-like traits to products using visual and verbal cues, significantly shaping consumers' impulse buying intentions. Visual cues, including facial expressions and body shapes, effectively capture attention and evoke positive emotions. The S-O-R model proposed by Kumar et al. (2020) [26] identifies visual anthropomorphic elements as external stimuli that shape emotional states and drive purchasing behavior. Such elements enhance product appeal and affinity, fostering emotional resonance and triggering immediate purchase tendencies. Visual anthropomorphism also enhances social presence, fostering a sense of social connection between consumers and the product [27]. This connection reduces perceived risk, builds trust, and encourages impulse buying. Verbal cues, including emotional language and humor, convey product personality and elicit empathetic responses. Cognitive appraisal theory Anita et al. (2023) [28] suggests that emotional language shapes attitudes and beliefs via deep cognitive processing, reinforcing product acceptance and trust. Verbal anthropomorphism imbues products with "personality," fostering emotional connections, amplifying positive emotions, lowering self-control, and boosting impulse purchases. Trust is a pivotal factor in both visual and verbal anthropomorphic design. According to Yturralde and Lazatin (2022) [29], consumers are more likely to trust products with human-like traits. Anthropomorphic design boosts product credibility, fostering a sense

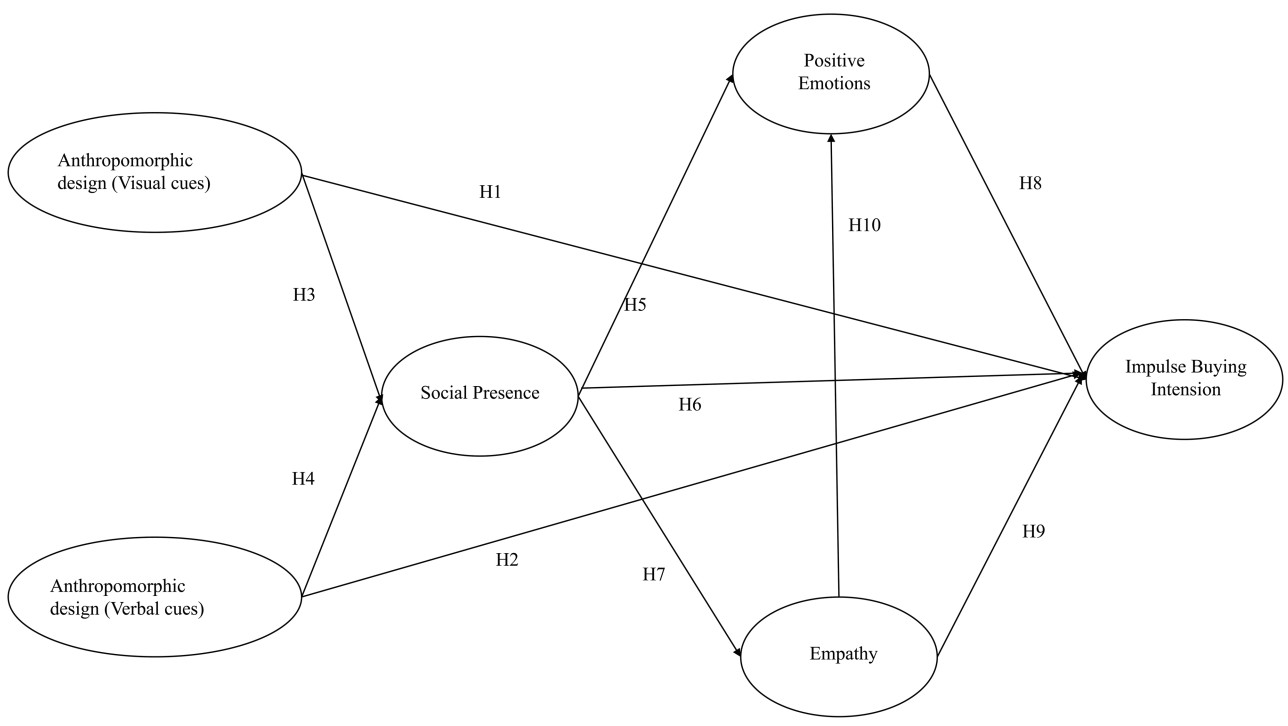

**Fig 1. Research model.**

of safety and prompting quicker purchasing decisions. By reducing uncertainty and perceived risk, trust facilitates immediate purchasing decisions. Visual cues impact emotions through sensory stimuli, while verbal cues influence attitudes via cognitive processing. Combined, these cues work synergistically to promote impulse buying behavior. Thus, the hypotheses formulated as follows.

H1: Anthropomorphic design in beverage packaging (visual cues) positively influences impulse buying intention.

H2: Anthropomorphic design in beverage packaging (verbal cues) positively influences impulse buying intention.

### 3.3 Anthropomorphic design in beverage packaging and social presence

Social presence is a well-studied concept in consumer behavior research. Zhang and Wang (2023) [30] suggest that it is a crucial factor in understanding consumer psychology. This is especially true in online and virtual interactions. Anthropomorphic beverage packaging enhances social presence by giving products human-like traits. Social presence is defined as the perception of others' presence and a sense of social connection during product interactions. Visual cues, such as facial expressions, body postures, and eye contact, mimic human appearance. These cues evoke consumers' perception of interpersonal interactions [31]. Visual anthropomorphism creates the impression of interacting with a real person. This strengthens social presence. Social response theory states that humans exhibit social responses to non-human objects displaying social cues. Visual anthropomorphic elements provide these cues. They allow the product to be perceived as a social entity. Andika et al. (2023) [32] argue that verbal cues, such as emotional language, anthropomorphic dialogue, and humor, convey the

product's "personality" and emotions. These verbal elements foster empathy and enhance a sense of connection . Media richness theory suggests that emotional and personalized language enhances information richness. This, in turn, strengthens social presence. Increased social presence fulfills consumers' social needs. This is particularly relevant for consumers experiencing loneliness [33]. It helps alleviate loneliness, improves mood, and triggers positive emotional responses. Enhanced social presence increases product likability and trust. This ultimately drives purchase intentions. Beverage packaging uses a variety of social cues through visual and verbal elements. These cues enhance social presence, meet social and emotional needs, and stimulate purchase intentions. Thus, the hypotheses formulated as follows.

H3: Anthropomorphic design in beverage packaging (visual cues) positively influences social presence.

H4: Anthropomorphic design in beverage packaging (verbal cues) positively influences social presence.

## 3.4 Social presence

Social presence, defined as the feeling of being with another person or entity in an intermediary environment, plays a crucial role in shaping consumer behavior [34] . When a product is designed with humanoid features, its social presence is enhanced, creating a sense of interaction and emotional connection [35]. Social presence can generate a sense of psychological intimacy, thereby enhancing consumers' emotional engagement with products or brands [36] . In the context of anthropomorphic design, humanoid features such as facial expressions or verbal cues can simulate social interactions, thereby generating warmth, pleasure, and satisfaction. Research has shown that social presence in both digital and physical environments can significantly enhance positive emotions. A higher level of social presence in the e-commerce environment will increase consumers' enjoyment and satisfaction. Similarly, social presence in website design has a positive impact on users' affective responses, such as trust and happiness. In addition, social presence enhances the perceived interactivity and emotional appeal of products, which may lead to impulsive purchasing behavior [37]. When consumers feel connected to a product, they are more likely to make unplanned purchases. When consumers perceive products to have qualities similar to humans, they are more likely to resonate with them, thereby strengthening their emotional connection. Thus, the hypotheses formulated as follows.

H5: Social presence positively influences positive emotions.

H6: Social presence positively influences impulse buying intention.

H7: Social presence positively influences empathy.

## 3.5 Positive emotions

Emotions result in favorable consumer evaluations, greater product likability, and stronger impulse buying intentions. Hedonic products particularly benefit from emotional and playful anthropomorphism. Based on the theory of affective responses , positive emotions enhance the attractiveness of products, create a sense of urgency, and make consumers more likely to make impulse purchases [38]. In addition, in the SOR model, stimuli such as product packaging or store environment can elicit affective responses (positive emotions) from consumers, which in turn drive behavioral responses (impulse buying) [39] . Positive emotions play a mediating role between external stimuli and impulsive buying behavior [40]. Therefore, anthropomorphic packaging can evoke consumer reactions and lead to the purchase of products. Rook et al. (2022) found that positive emotions such as excitement and happiness significantly increase the likelihood of impulse buying [41]. Their research emphasizes that

consumers who experience positive emotions are more likely to take action on spontaneous desires without deep consideration. Herabadi explored the role of emotional states in impulse buying and found that positive emotions significantly increased impulse buying tendencies. They believe that positive emotions can lower consumers' self-control and make them more likely to make impulsive decisions. Zhang et al. studied the impact of positive emotions on impulse buying in online shopping contexts. Their research findings indicate that positive emotions, such as happiness and satisfaction, significantly enhance impulse buying intentions by increasing consumers' perceived enjoyment and reducing their hesitation. Hence, the following hypothesis is constructed.

H8: Positive Emotions positively influences impulse buying intention.

### 3.6 Empathy

Empathy plays a pivotal role in consumer decision-making, particularly when objects express emotions that evoke emotional resonance. Visual anthropomorphic elements, which mimic human emotions, lead consumers to perceive the product as an emotional entity [42]. According to linguistic empathy theory, language that conveys emotions and personality traits can further enhance empathy. Emotional language helps consumers understand and relate to the emotions expressed by the product, eliciting empathetic responses [43]. High levels of empathy strengthen emotional bonds between consumers and products, fostering greater brand loyalty and purchase intentions. By eliciting empathy, anthropomorphic design enhances the product's emotional appeal and distinctiveness, fostering affinity and trust, which in turn shape purchasing behavior. Positive anthropomorphic cues in beverage packaging evoke state empathy, enhancing consumers' impulse buying intentions and amplifying positive emotions. Empathy not only influences impulse buying intentions but also plays a crucial role in evoking positive emotions such as happiness, satisfaction, and joy [44]. These positive emotions enhance the overall appeal of the product and drive consumer behavior. Empirical studies support these relationships. For instance, Su et al. (2023) found that empathy significantly mediates the relationship between anthropomorphic design and purchase intention [45]. Their research suggests that consumers with stronger empathy toward anthropomorphic products are more likely to make unplanned purchases. Similarly, Rachmad (2024) emphasized that empathy-driven emotional resonance enhances brand loyalty and impulse buying behavior, particularly in emotionally charged situations [46]. These findings highlight the dual role of empathy in both driving impulse buying intentions and evoking positive emotions, which collectively enhance the product's appeal and influence consumer behavior. Thus, the hypotheses formulated as follows.

H9: Empathy positively influences impulse buying intention.

H10: Empathy positively influences positive emotions.

## 4 Experimental results

### 4.1 Questionnaire source

The questionnaire for this study aims to investigate the effects of anthropomorphic beverage packaging design on impulse buying intentions and its underlying mechanisms. The questionnaire is divided into third sections: design elements, consumer emotional responses, individual psychological traits, and purchase behavior. The first section covers Anthropomorphic Design, specifically focusing on visual and verbal cues. The second section addresses Positive Emotions, Social Presence, and Empathy. The third focuses on Impulse Buying Intention. The questionnaire is designed with a rational structure, ensuring comprehensive coverage

of all variables relevant to the study. It maintains logical relationships and a clear hierarchical structure among variables. Questions in each section were adapted from classic scales in related fields and tailored to the context of anthropomorphic beverage packaging design. All measurement items were sourced from established scales in related fields and were tailored to the specific context of anthropomorphic beverage packaging design, as shown in Table 1. Data collection occurred between June 1, 2024, and September 30, 2024. The survey targeted beverage consumers in Shenzhen, Guangdong Province. A total of 308 valid responses were collected. All participants provided informed consent and adhered to the Ethics Committee's requirements, ensuring data authenticity and research compliance.

## 4.2 Analysis of basic information from the questionnaire

This study's questionnaire is designed to investigate how anthropomorphic beverage packaging influences consumers' impulsive buying intentions and the mechanisms underlying this effect. The questionnaire employed detailed variables and structured questions to gather comprehensive data on consumer demographics, purchasing behavior, and psychological states. Fig 2 shows that 42.9% of respondents are aged 24 to 30, representing the largest age group and the primary market segment. The second-largest group (19–23 years) comprises 26.9% of respondents, followed by the 31–35 age group at 18.5%. Respondents under 18 and those aged 36–45 form smaller groups, at 6.2% and 5.5%, respectively. These findings highlight young and middle-aged consumers as the primary targets in the beverage market. Regarding income, 45.6% of respondents earn 4,001–6,000 RMB monthly, forming the largest group. The next largest group (25.7%) earns 6,001–8,000 RMB monthly. Those earning less than 4,000 RMB and over 8,000 RMB represent 20.8% and 7.9% of respondents, respectively. These results suggest that middle-income consumers constitute the core market segment. Regarding monthly expenditure, 59.4% of respondents spend 501–1,000 RMB, representing the largest group. The second-largest group (18.8%) spends 1,001–1,500 RMB. Respondents spending 1,501–2,000 RMB and over 2,001 RMB account for 8.4% and 7.1%, respectively. Just 6.2% of respondents report a monthly expenditure below 500 RMB. These findings reveal that mid-to-low income groups drive the beverage market's spending power. This pattern aligns with the classification of beverages as fast-moving consumer goods. Concerning packaging preferences, vintage and classic designs are the most favored, with 51.9% of respondents indicating a preference for these styles. Unique and innovative styles rank second, preferred by 39.0% of respondents. Minimalist and modern designs attract 12.9% of respondents. The least preferred styles are eco-friendly and natural (4.9%) and bright and colorful (3.2%). These findings suggest a preference for packaging that conveys cultural value and emotional appeal.

**Table 1. Sources of questionnaire measurement items.**

| Variables | Source |
|---|---|
| Anthropomorphic design (Visual cues) | Aggarwal & McGill (2007) [47]; Guido & Peluso (2015) [48]; Mourey et al. (2017) [49]; Kim & McGill (2018) [50] |
| Anthropomorphic design (Verbal cues) | Bartneck et al. (2009) [51]; Chen & Ham (2024) [52] |
| Impulse Buying Intention | Beatty & Ferrell (1998) [53]; Park et al. (2012) [54] |
| Positive Emotions | Hsieh et al. (2021) [55]; Mummalaneni (2005) [56]; Mehrabian & Russell (1974) [57] |
| Social Presence | Gefen & Straub (2003) [58]; Lim & Soutar (2021)[59]; Gao & Li (2017)[60] |
| Empathy | Escalas & Stern (2003) [61]; Shen (2010) [62] |

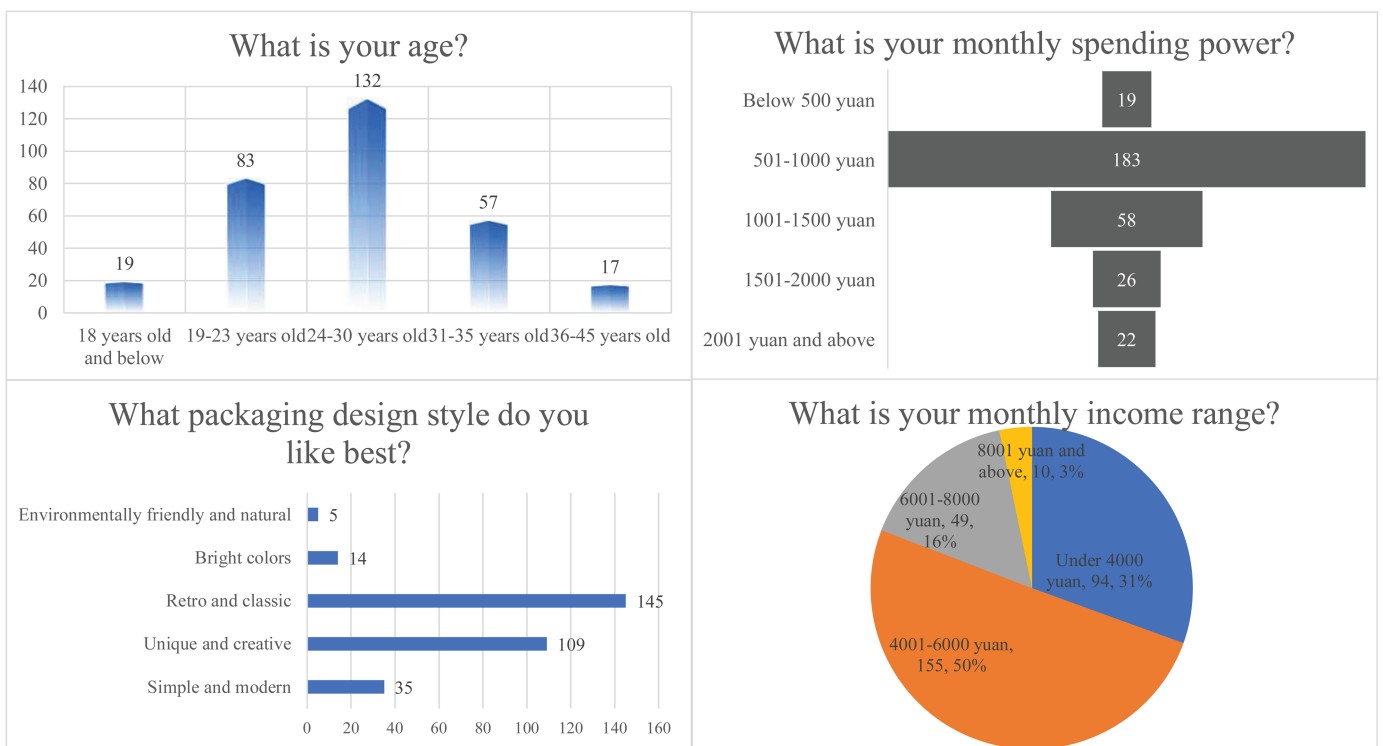

**Fig 2. Basic information distribution statistics.**

## 4.3 Exploratory factor analysis

**4.3.1 Reliability and validity analysis.** This study employed exploratory factor analysis (EFA) to assess the construct validity and internal consistency of the questionnaire. Cronbach's Alpha coefficient was first applied to examine the internal consistency of each construct in the questionnaire. Table 2 indicates that all variables have Alpha values exceeding 0.8, with an overall Cronbach's Alpha of 0.936. These findings demonstrate high internal consistency across all constructs. Next, the Kaiser-Meyer-Olkin (KMO) measure and Bartlett's sphericity test were employed to assess the sample's suitability for factor analysis. The KMO statistic measures the adequacy of the sample for factor analysis. All variables exhibit KMO values exceeding 0.7, with an overall KMO of 0.935, significantly surpassing the minimum threshold of 0.6. These results confirm the data's suitability for factor analysis. Furthermore, the results validate the questionnaire's high reliability and validity in assessing the effects of anthropomorphic beverage packaging on consumers' impulse buying intentions and related psychological mechanisms.

**4.3.2 Factor number analysis.** This study performed a factor analysis on the questionnaire data to identify the optimal number of factors. Using Kaiser's criterion (eigenvalue > 1) and cumulative explained variance, five primary factors were extracted, as presented in Table 3. The initial eigenvalues of the five factors were 10.111, 2.858, 2.084, 1.463 and 1.009, all exceeding 1, satisfying Kaiser's extraction criterion. Together, these factors explained 67.405% of the total variance, sufficiently capturing the main variability in the data. A detailed analysis of factor composition revealed: the first factor comprised Anthropomorphic Design (Verbal Cues), Impulse Buying Intention, and Positive Emotions. The second factor included only

**Table 2. Reliability and effectiveness.**

| Variables | Items | Alpha | KMO |
|---|---|---|---|
| Anthropomorphic design (Visual cues) | 4 | .887 | .841 |
| Anthropomorphic design (Verbal cues) | 4 | .849 | .813 |
| Impulse Buying Intention | 4 | .863 | .821 |
| Positive Emotions | 6 | .892 | .902 |
| Social Presence | 4 | .880 | .837 |
| Empathy | 4 | .858 | .820 |
| **Total** | **26** | **.936** | **.935** |

**Table 3. Results of factor number analysis.**

| Item | Initial Eigenvalues | | | Rotated Loadings Sum of Squares | | |
|---|---|---|---|---|---|---|
| | Eigenvalue | Variance (%) | Cumulative (%) | Sum of Squares | Variance (%) | Cumulative (%) |
| 1 | 10.111 | 38.889 | 38.889 | 4.216 | 16.215 | 16.215 |
| 2 | 2.858 | 10.992 | 49.882 | 3.747 | 14.411 | 30.626 |
| 3 | 2.084 | 8.015 | 57.897 | 3.256 | 12.523 | 43.149 |
| 4 | 1.463 | 5.626 | 63.522 | 3.184 | 12.246 | 55.395 |
| 5 | 1.009 | 3.882 | 67.405 | 3.123 | 12.010 | 67.405 |

Anthropomorphic Design (Visual Cues). The third, fourth, and fifth factors corresponded to Social Presence, Empathy, and Loneliness, respectively. After rotation, squared loadings exceeded 0.7 for all variables within each factor. These high loadings confirmed the rationality of the factor structure and demonstrated convergent validity, ensuring independence and validity for each factor. Furthermore, low correlations between the five factors confirmed their independence. This finding supports the model's simplicity and explanatory power.

## 4.4 Factor analysis

Factor analysis results in Table 4 show that five main factors were extracted: anthropomorphic design, social presence, impulse buying intention, positive emotions, and empathy. In anthropomorphic design, the ADVEC series indicators had high loadings (approximately .715 to .755). This indicates that consumers are sensitive to emotional language. The ADVIC series indicators had lower loadings (approximately .428 to .541). This reflects that the role of visual cues is more dispersed. The loadings for the social presence factor ranged from .547 to .572. This indicates that consumers have consistent perceptions of the social interaction conveyed by the packaging. The impulse buying intention factor indicators all exceeded .800. This shows high internal consistency and provides empirical support for the direct influence of external stimuli on purchasing behavior. The indicators for both positive emotions and empathy had loadings above .763 and .725, respectively. This verifies the robustness of emotional responses and empathy experiences. The high loadings for each factor validate the convergent validity and internal consistency of the measurement model. They also suggest that consumers place greater importance on emotional language information. This provides a solid foundation for model refinement, theoretical expansion, and practical application.

## 4.5 Confirmatory factor analysis

**4.5.1 Model fit analysis.** To validate the questionnaire's structural validity, confirmatory factor analysis (CFA) was conducted to assess model fit. Table 5 provides the CFA model fit indices. The chi-square value (CMIN) was 304.619, with degrees of freedom (DF) of 284. The

**Table 4. Factor components.**

| Construct | | Component 1 | Component 2 | Component 3 | Component 4 | Component 5 |
|---|---|---|---|---|---|---|
| Anthropomorphic Design | ADVEC2 | .755 | | | | |
| | ADVEC3 | .755 | | | | |
| | ADVEC4 | .731 | | | | |
| | ADVEC1 | .715 | | | | |
| | ADVIC2 | .541 | | | | |
| | ADVIC4 | .517 | | | | |
| | ADVIC3 | .513 | | | | |
| | ADVIC1 | .428 | | | | |
| Social Presence | SP4 | | .572 | | | |
| | SP2 | | .563 | | | |
| | SP3 | | .548 | | | |
| | SP1 | | .547 | | | |
| Impulse Buying Intension | IBI4 | | | .819 | | |
| | IBI1 | | | .809 | | |
| | IBI2 | | | .808 | | |
| | IBI3 | | | .800 | | |
| Positive Emotions | PE1 | | | | .786 | |
| | PE5 | | | | .786 | |
| | PE6 | | | | .782 | |
| | PE2 | | | | .781 | |
| | PE3 | | | | .767 | |
| | PE4 | | | | .763 | |
| Empathy | EM4 | | | | | .827 |
| | EM3 | | | | | .782 |
| | EM1 | | | | | .771 |
| | EM2 | | | | | .725 |

**Table 5. Model fit indices for confirmatory factor analysis.**

| Model Fit | CMIN | DF | CMIN/DF | NFI | RFI | IFI | TLI | CFI | GFI | RMSEA |
|---|---|---|---|---|---|---|---|---|---|---|
| Fit Results | 304.619 | 284 | 1.073 | .938 | .929 | .996 | .995 | .995 | .930 | .015 |
| Judgment Std. | - | - | <3 | >0.9 | >0.9 | >0.9 | >0.9 | >0.9 | >0.9 | <0.08 |

CMIN/DF ratio was 1.073, below the threshold of 3, indicating a good model fit. Fit indices included NFI (0.938), RFI (0.929), IFI (0. 996), TLI (0.995), CFI (0.995), and GFI (0. 930), all exceeding the standard threshold of 0.9. These indices confirmed excellent model fit. Additionally, the RMSEA (Root Mean Square Error of Approximation) was 0.015, below 0.08, indicating low error and an ideal fit. All fit indices met or exceeded the recommended thresholds. These results demonstrate that the constructed measurement model possesses strong statistical fit and explanatory power.

**4.5.2 Convergent validity and discriminant validity analysis.** This study performed convergent and discriminant validity analyses to verify the structural validity of the questionnaire measurement model. Table 6 shows that all constructs have factor loadings exceeding 0.7. This indicates strong correlations between measurement items and their corresponding latent variables, confirming convergent validity. Additionally, the composite reliability (CR) values for all constructs exceeded 0.7, meeting the recommended threshold. These results further support the questionnaire's internal consistency and measurement reliability. All constructs demonstrated AVE (average variance extracted) values exceeding 0.5. This satisfies the standard for convergent validity, indicating that each construct effectively explains the variance of its measurement items. For discriminant validity, Table 7 shows the correlation coefficients between constructs. These coefficients were compared with the square roots of

**Table 6. Convergent validity and composite reliability.**

| Construct | Item | Loading Factor | CR | AVE |
|---|---|---|---|---|
| Anthropomorphic design (Visual cues) | ADVI4 | 0.836 | 0.888 | 0.664 |
| | ADVI3 | 0.796 | | |
| | ADVI2 | 0.840 | | |
| | ADVI1 | 0.787 | | |
| Anthropomorphic design (Verbal cues) | ADVE4 | 0.803 | 0.849 | 0.586 |
| | ADVE3 | 0.815 | | |
| | ADVE2 | 0.731 | | |
| | ADVE1 | 0.707 | | |
| Positive Emotions | PM11 | 0.774 | 0.892 | 0.579 |
| | PM12 | 0.754 | | |
| | PM13 | 0.759 | | |
| | PM14 | 0.757 | | |
| | PM15 | 0.755 | | |
| | PM16 | 0.770 | | |
| Social Presence | SP21 | 0.845 | 0.880 | 0.648 |
| | SP22 | 0.779 | | |
| | SP23 | 0.771 | | |
| | SP24 | 0.822 | | |
| Empathy | EM34 | 0.825 | 0.859 | 0.605 |
| | EM33 | 0.798 | | |
| | EM32 | 0.739 | | |
| | EM31 | 0.746 | | |
| Impulse Buying Intention | IBI1 | 0.750 | 0.864 | 0.614 |
| | IBI2 | 0.825 | | |
| | IBI3 | 0.770 | | |
| | IBI4 | 0.787 | | |

**Table 7. Discriminant validity.**

| | ADVI | ADVE | PM | SP | EM | IBI |
|---|---|---|---|---|---|---|
| Anthropomorphic design (Visual cues) | 0.815 | | | | | |
| Anthropomorphic design (Verbal cues) | 0.467 | 0.765 | | | | |
| Positive Emotions | 0.298 | 0.401 | 0.761 | | | |
| Social Presence | 0.693 | 0.677 | 0.419 | 0.805 | | |
| Empathy | 0.340 | 0.555 | 0.410 | 0.475 | 0.778 | |
| Impulse Buying Intention | 0.610 | 0.743 | 0.564 | 0.769 | 0.640 | 0.783 |

AVE values from Table 7. The AVE values for all constructs exceeded their correlations with other constructs. These findings confirm strong discriminant validity, ensuring the measurement model's independence and accuracy.

## 4.6 Structural equation model path analysis

This study employed a structural equation model to analyze each path coefficient in detail. The model fit was good. The theoretical constructs and empirical data were highly consistent. The results (see Table 8) show that visual cues in packaging design significantly enhanced social presence ($\beta$ = 0.506, SE = 0.061, C.R. = 8.308, p < 0.001). Verbal cues also significantly enhanced social presence ($\beta$ = 0.510, SE = 0.063, C.R. = 8.119, p < 0.001). These findings indicate that consumers experienced clear social interaction when perceiving anthropomorphic features. Further analysis revealed that social presence significantly influenced empathy ($\beta$ = 0.507, SE = 0.064, C.R. = 7.940, p < 0.001) and positive emotions ($\beta$ = 0.276, SE = 0.064, C.R. = 4.287, p < 0.001). Empathy also exerted a positive effect on positive emotions ($\beta$ = 0.230,

**Table 8. Path coefficients in the structural equation model.**

| Hyp. | Path | Estimate | S.E. | C.R. | P | Conclusion |
|------|------|----------|------|------|---|------------|
| H3 | Social Presence ← Anthropomorphic design (Visual cues) | 0.506 | 0.061 | 8.308 | ∗∗∗ | Supported |
| H4 | Social Presence ← Anthropomorphic design (Verbal cues) | 0.510 | 0.063 | 8.119 | ∗∗∗ | Supported |
| H7 | Empathy ← Social Presence | 0.507 | 0.064 | 7.940 | ∗∗∗ | Supported |
| H5 | Positive Emotions ← Social Presence | 0.276 | 0.064 | 4.287 | ∗∗∗ | Supported |
| H10 | Positive Emotions ← Empathy | 0.230 | 0.065 | 3.555 | ∗∗∗ | Supported |
| H1 | Impulse Buying Intension ← Anthropomorphic design (Visual cues) | 0.130 | 0.056 | 2.299 | 0.022 | Supported |
| H2 | Impulse Buying Intension ← Anthropomorphic design (Verbal cues) | 0.258 | 0.060 | 4.285 | ∗∗∗ | Supported |
| H8 | Impulse Buying Intension ← Positive Emotions | 0.190 | 0.046 | 4.140 | ∗∗∗ | Supported |
| H6 | Impulse Buying Intension ← Social Presence | 0.005 | 0.087 | 2.264 | 0.101 | Not Supported |
| H9 | Impulse Buying Intension ← Empathy | 0.189 | 0.044 | 4.284 | ∗∗∗ | Supported |

SE = 0.065, C.R. = 3.555, p < 0.001). In terms of impulse buying intention, the direct effect of visual cues was weak ($\beta$ = 0.130, SE = 0.056, C.R. = 2.299, p = 0.022), whereas verbal cues had a strong direct influence ($\beta$ = 0.258, SE = 0.060, C.R. = 4.285, p < 0.001). In addition, positive emotions ($\beta$ = 0.190, SE = 0.046, C.R. = 4.140, p < 0.001) and empathy ($\beta$ = 0.189, SE = 0.044, C.R. = 4.284, p < 0.001) significantly promoted impulse buying intention. The direct effect of social presence on impulse buying intention was not significant ($\beta$ = 0.005, SE = 0.087, C.R. = 2.264, p=0.101), which suggests that social presence mainly exerts its influence through the mediating roles of empathy and positive emotions. Overall, the visual and verbal cues in anthropomorphic packaging design enhance social presence, which in turn indirectly stimulates empathy and positive emotions, thereby driving impulse buying intention, with the direct effects of verbal cues and emotional factors being particularly significant.

This study further assessed potential common method bias using the ULMC approach (see Table 9). The results confirmed that CMV did not significantly influence our path coefficients, reinforcing the robustness of the hypothesized relationships.The model fitting index shows that compared with the original model, the ULMC model did not significantly improve the fitting degree (ΔCFI = 0.002; ΔRMSEA = 0.001) . The single factor model showed poor fit (CFI = 0.72, RMSEA = 0.14), confirming that CMV is unlikely to confuse our results. Therefore, the fitting index of the ULMC model did not significantly improve, and the poor fitting of the single factor model indicates that the variance of commonly used methods did not substantially affect our structural equation modeling results.

### 4.7 Discussion

The empirical results of this study generally align with theoretical expectations (see Fig 3). The findings reveal some subtle differences. Visual cues and verbal cues both effectively enhance consumers' sense of social presence, which supports social presence theory. Products simulate interpersonal interaction to provide a context for emotional experiences and subsequent behavior. Although theory predicts that social presence will trigger empathy, the revised

**Table 9. ULMC analysis for common method variance.**

| Model | $\chi^2$/df | CFI | RMSEA | SRMR |
|-------|-------------|-----|-------|------|
| Original Measurement Model | 1.92 | 0.97 | 0.04 | 0.03 |
| ULMC Model | 1.89 | 0.97 | 0.04 | 0.03 |
| Single-Factor Model | 4.76 | 0.72 | 0.14 | 0.12 |

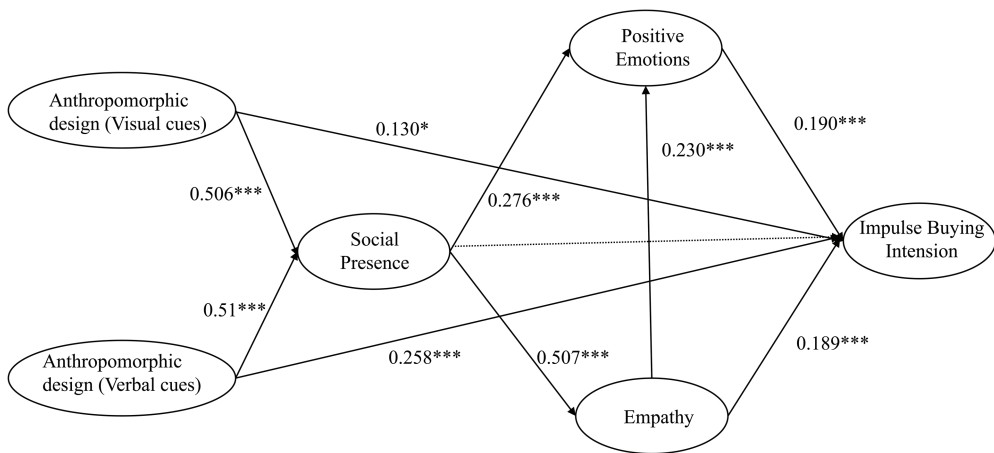

**Fig 3. Path coefficients for hypotheses.**

empirical results confirm a strong direct relationship between them, indicating that a heightened sense of social presence is indeed a critical antecedent to evoking empathy. In addition, positive emotions are influenced by both social presence and empathy. This is consistent with the stimulus-organism-response model and emotional response theory, and it further demonstrates that positive emotions mediate impulse buying intention. Notably, while empathy and positive emotions play significant roles in shaping impulse buying behavior, the direct effect of social presence on impulse buying intention is not statistically significant. In summary, the results indicate that anthropomorphic design significantly influences consumer impulse buying behavior through a series of interconnected psychological processes. These findings provide theoretical support and practical insights for packaging design and marketing strategies, while also suggesting promising directions for further exploration of mechanisms that activate consumer empathy.

By synthesizing the statistical results of all hypotheses, we find a close relationship between the p-values and the path coefficients. This reflects both the statistical significance of the relationships among variables and their practical impact within the theoretical model. Low p-values indicate highly significant relationships, while larger path coefficients suggest stronger effects in real contexts. This finding is consistent with Chitturi et al. (2021) on the impact of emotional and visual cues in packaging design on consumer behavior [63]. At the same time, some paths, though statistically significant, have smaller coefficients, suggesting that companies should focus on factors with larger effects or consider combining additional design strategies to amplify the weaker ones. This aligns with Kumar et al. (2023) regarding how language and visual cues in anthropomorphic design evoke emotional resonance [64]. Furthermore, the model validates the mediating roles of social presence and positive emotions in consumer decision-making. This provides theoretical support for companies to enhance social interaction and emotional experience in packaging design, and it demonstrates that reinforcing these factors can indirectly boost impulse buying intention, which is consistent with LI et al. (2023) [65]. In summary, these internal relationships not only provide strong support for the theoretical model but also offer important insights for practical marketing strategies. Companies should prioritize factors that are both statistically significant and have substantial effects by reinforcing emotional language and building a strong sense of social presence to ultimately maximize positive emotions and impulse buying intention.

The findings of this study not only validate the effectiveness of the theoretical model but also offer feasible insights for innovations in packaging design and marketing strategies. The results indicate that, although most hypotheses were supported, the direct effect of social presence on impulse buying intention was not significant. This suggests that relying solely on creating a social atmosphere is insufficient to directly evoke consumers' impulse buying. Instead, social presence primarily operates by enhancing empathy and positive emotions. Therefore, companies should employ multiple methods during the design process. For example, they should incorporate emotional expression and storytelling into both visual and verbal design and leverage digital interactive technologies such as augmented reality and virtual experiences to create immersive consumption scenarios. This approach can effectively enhance social presence while more robustly triggering empathy and positive emotions. Cheng et al. (2022) pointed out that high-quality emotional and visual elements can significantly enhance consumers' emotional identification with a product [66]. Prisca et al. (2023) emphasized that anthropomorphic design makes a product more appealing and narrows the emotional distance between consumers and the brand [67]. Zhang et al. (2023) confirmed that immersive experiences created by emerging interactive technologies help strengthen social interaction and emotional resonance, thereby increasing purchase intention [68]. These empirical results provide companies with a clear strategic direction. In a competitive market, companies should prioritize design factors that are both statistically significant and have large effects, enhancing brand attractiveness and competitiveness through innovative, multi-dimensional, and multi-sensory strategies. Simultaneously, they should pay attention to the synergistic effects among various factors by integrating emotional language, rich visual presentation, and interactive experiences to build a comprehensive and emotionally engaging marketing ecosystem that meets diverse consumer needs. It is important to note that this study has limitations regarding the sample region and consumer group as well as in the measurement and interpretation of the empathy mechanism. Future research can introduce additional contextual variables and longitudinal data to further explore the dynamic relationships among the factors in anthropomorphic design, thereby providing more solid support for theoretical expansion and practical application.

## 5 Implications

### 5.1 Theoretical implications

This study employs empirical analysis to explore how visual and verbal cues in anthropomorphic beverage packaging design influence consumers' impulse buying intentions. It contributes to the theoretical understanding of anthropomorphic design and consumer behavior by validating the applicability of the Stimulus-Organism-Response (S-O-R) model. The findings demonstrate that external stimuli (anthropomorphic design) influence behavioral responses (impulse buying intention) by shaping internal psychological states, including positive emotions, social presence, and empathy. This mechanism is empirically supported [69]. The study underscores the distinct roles of visual and verbal cues in anthropomorphic design. Visual cues primarily enhance positive emotions and social presence, while verbal cues evoke stronger empathetic responses. These findings provide deeper insights into the mechanisms underlying different types of cues in anthropomorphic design. They also align with Lim et al. (2021)'s [70] cognitive appraisal theory, which posits that verbal information induces deeper cognitive processing and attitude changes. Furthermore, the study reveals that visual anthropomorphic design does not significantly affect empathy, suggesting that verbal cues are more effective in eliciting deep emotional resonance. This offers new perspectives for anthropomorphic design theory. By integrating theories of anthropomorphic

design, positive emotions, social presence, and empathy, this study develops a systematic theoretical model. It explains the psychological mechanisms through which anthropomorphic design in beverage packaging influences impulse buying intentions, thereby advancing research in related fields.

## 5.2 Practical implications

This study provides valuable insights for businesses in beverage packaging design and marketing strategies. First, verbal anthropomorphic design plays a more significant role than visual design in stimulating consumers' impulse buying intentions. It evokes positive emotions, enhances social presence, and triggers empathy. This finding aligns with Huang et al. (2023) [71] cognitive appraisal theory, which suggests that detailed verbal information induces deeper cognitive processing and attitude changes. Therefore, businesses should incorporate emotional language, humor, and personalized expressions into packaging to convey product emotions and personality, thereby influencing consumer decisions. Since visual design does not significantly impact empathy, combining visual and verbal elements can create multidimensional and vivid product images. For example, pairing anthropomorphic cartoon characters with emotional statements on packaging can address the limitations of visual cues and enhance marketing effectiveness [72]. Additionally, businesses should leverage social media's interactivity and reach to amplify the impact of anthropomorphic design. By integrating online and offline strategies, companies can share anthropomorphic product images and stories to enhance consumer engagement and build a sense of community. Finally, companies should continuously monitor consumer behavior and market trends to dynamically adjust their strategies. They should innovate anthropomorphic designs to cater to diverse cultural needs, ensuring effectiveness and competitiveness in the global market.

## 6 Conclusion

This study investigated how anthropomorphic design in beverage packaging affects consumers' impulse buying intention. The research results indicate that both visual and verbal cues can significantly enhance impulse buying intention. This study validated the stimulus biological response model and emotion theory in the context of anthropomorphic design, enriching the relevant theoretical framework. The research results indicate that social presence has no significant impact on impulse buying. In the context of beverage packaging, consumers may be more concerned with product functionality (such as taste, thirst quenching) or visual appeal rather than social interaction or emotional connection. Social presence is usually more important in products that are highly social or interactive, such as social media platforms or virtual assistants, while its impact may be weaker in low engagement products like beverages. In summary, this study encourages companies to use emotional language, humor, and personalized expression in their packaging. Combining visual and linguistic personification elements can create more attractive product images. The use of social media can expand the influence of anthropomorphic design and increase consumer engagement. However, this study is limited by its regional sample size. Future research should expand the sample size and geographical scope. This study only focuses on visual and verbal cues; Future research can explore multi sensory anthropomorphic design. In addition, cross-sectional design limits the ability to analyze long-term effects. Future research should adopt a longitudinal approach to examine long-term effects and provide deeper insights.

## Questionnaire

The questionnaire is shown in Table 10.

**Table 10. Questionnaire information.**

| | |
|---|---|
| **Anthropomorphic Design (Visual Cues)** | 1 – The product (packaging) looks like a person.<br>2 – The product (packaging) has a human-like expression.<br>3 – The product (packaging) has a human-like appearance.<br>4 – The product (packaging) has come alive. |
| **Anthropomorphic Design (Verbal Cues)** | 1 – The verbal style of this product (packaging) is natural.<br>2 – The verbal style of this product (packaging) is humanlike.<br>3 – The verbal style of this product (packaging) is conscious.<br>4 – The verbal style of this product (packaging) is elegant. |
| **Impulse Buying Intention** | 1 – When I saw the product (packaging), I had a strong desire to buy it.<br>2 – When I saw the product (packaging), I wanted to have it immediately.<br>3 – When I saw the product (packaging), I felt that the product was what I wanted.<br>4 – When I saw the product (packaging), I wanted to buy it even though it was not what I had planned to buy before. |
| **Positive Emotions (Pleasure & Arousal)** | 1 – When I saw the product (packaging), I felt happy.<br>2 – When I saw the product (packaging), I felt pleasure.<br>3 – When I saw the product (packaging), I felt hopeful.<br>4 – When I saw the product (packaging), I felt satisfied.<br>5 – When I saw the product (packaging), I felt active.<br>6 – When I saw the product (packaging), I felt excited. |
| **Social Presence** | 1 – There is a sense of human contact with this product (packaging).<br>2 – This product (packaging) has a personalized touch.<br>3 – There is a human-like warmth associated with this product (packaging).<br>4 – There is a sense of human sensitivity associated with this product (packaging). |
| **Empathy** | 1 – While watching the product (packaging), I can feel the packaging's emotions.<br>2 – While watching the product (packaging), I experienced many of the same feelings that the packaging portrayed.<br>3 – While watching the product (packaging), I can understand what the packaging is going through.<br>4 – While watching the product (packaging), I felt like I was the packaging. |

# 7 Supporting information

**S1 Appendix. Further details regarding the anthropomorphic design of this study.**
(PDF)

# Author contributions

**Conceptualization:** Du Yinghui, Zenghui Tian.

**Data curation:** Yuyang Tian, Du Yinghui, Zenghui Tian.

**Formal analysis:** Zenghui Tian.

**Investigation:** Ye Qiu.

**Methodology:** Siti Farhana Zakaria.

**Supervision:** Siti Farhana Zakaria.

**Validation:** Siti Farhana Zakaria.

**Visualization:** Ye Qiu.

**Writing – original draft:** Yuyang Tian, Siti Farhana Zakaria, Zenghui Tian.

**Writing – review & editing:** Yuyang Tian, Du Yinghui, Ye Qiu.

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
