## [Decision Letter · Decision Letter 0]

11 Feb 2025

PONE-D-24-59115Can Anthropomorphic Design in Beverage Packaging Enhance Impulse Buying Intention? Amazing Visual and Verbal Cues!PLOS ONE

Dear Dr. Tian,

Thank you for submitting your manuscript to PLOS ONE. After careful consideration, we feel that it has merit but does not fully meet PLOS ONE’s publication criteria as it currently stands. Therefore, we invite you to submit a revised version of the manuscript that addresses the points raised during the review process.

Editor's comments:

Since our expert reviewers recognize the merits of your work, I am willing to provide an opportunity for revision. Please carefully consider the reviewers' comments and address each concern systematically and in detail.

We look forward to receiving your revised manuscript.

Kind regards,

Simon Dang

Academic Editor

PLOS ONE

4. Please provide the Questionnaires used in this study.

5. We note that Appendix includes an image of a participant in the study.

As per the PLOS ONE policy (http://journals.plos.org/plosone/s/submission-guidelines#loc-human-subjects-research) on papers that include identifying, or potentially identifying, information, the individual(s) or parent(s)/guardian(s) must be informed of the terms of the PLOS open-access (CC-BY) license and provide specific permission for publication of these details under the terms of this license. Please download the Consent Form for Publication in a PLOS Journal (http://journals.plos.org/plosone/s/file?id=8ce6/plos-consent-form-english.pdf). The signed consent form should not be submitted with the manuscript, but should be securely filed in the individual's case notes. Please amend the methods section and ethics statement of the manuscript to explicitly state that the participant has provided consent for publication: “The individual in this manuscript has given written informed consent (as outlined in PLOS consent form) to publish these case details”.

Reviewers' comments:

Reviewer's Responses to Questions

**Comments to the Author**

1. Is the manuscript technically sound, and do the data support the conclusions?

Reviewer #1: Partly

Reviewer #2: Partly

2. Has the statistical analysis been performed appropriately and rigorously? 

Reviewer #1: No

Reviewer #2: Yes

3. Have the authors made all data underlying the findings in their manuscript fully available?

Reviewer #1: Yes

Reviewer #2: Yes

4. Is the manuscript presented in an intelligible fashion and written in standard English?

Reviewer #1: Yes

Reviewer #2: Yes

5. Review Comments to the Author

Reviewer #1: 1. This research contains too many simple and isolated sentences, which might cause boredom and confusions to the readers. Below are just examples. The author should proofread the whole manuscript:

“…This involves assigning human traits to non-human objects. Such design creates human-like appearances or behaviors. It aims to evoke emotional resonance in consumers. This strategy is widely used in marketing. It boosts product appeal and fosters consumer engagement….”

“Trust is another key mechanism. It reduces perceived risk. Trust enhances product credibility and reliability. This increases consumer trust and purchase likelihood. Empathy is defined as the ability to understand and share others’ emotions”

2. It is unclear about the context of this study. Is the impulse buying intention in a physical setting/m-commerce/website/banner ads, etc.? This needs to be explained in more detailed.

3. Line 61: anthropomorphic packaging was mentioned, however, it was not explicitly explained what is meant by anthropomorphic packaging. In addition, there are many weak supporting evidence. For example, it was stated that “sales surged by 4,600%”, is it merely due to anthropomorphic packaging? Are there any other factors, e.g. high brand awareness due to investment in ads, or promotion?. If the “surge” is in conjunction with many other factors, then these evidence are not strong enough to conclude that “These data suggest that anthropomorphic design in packaging significantly enhances consumers’ emotional identification and purchase intentions”

4. Line 74: There are many weak arguments, for example “whether anthropomorphic language styles influence consumer behavior through psychological mechanisms such as social presence, positive emotions, and empathy remains insufficiently explored.” Actually, there have been many research on anthropomorphic language styles and social presence. There are just a few references in the introduction.

5. Line 122: The author stated that “Anthropomorphic beverage packaging, which assigns human traits to products…”. However, the first hyperlink (DIGITALING) in line 61 shows photos of fruits (e.g. banana and strawberry) and animals (e.g. cat). The definition is not actually consistent to examples given. This will cause confusions to readers.

6. Line 164: The author has immediately switched from Concept of Impulse Buying Intention to SOR. It is important to create a reading flow for readers. In addition, the author needs to justify how SOR framework is applicable in the context of Impulse Buying Intention. Furthermore, SOR model should be explained in greater details.

7. In regards with questionnaires used in this research, all questions for Social presence are about human-likeness, which is the same as anthropomorphism. Therefore, it is very difficult to come up with a sound analysis. In addition, in table 1, sources of questionnaire measurement items were provided, however, it is unclear which one the author has adapted because the author has provided at least 2 sources for 1 construct. Take a look at reference 64, 5 items for social presence are much different from the items used in this research. This might be due to poor application of back-translating technique, leading to inaccurate analyses and findings.

8. Line 454: it is unclear if the author conducted an experimental research, or a Covariance-Based Structural Equation Modeling.

9. Marker variable test for common method bias check was missing.

10. Headings were not appropriately deployed, for example, Line 847

11. Control variables in the research model and analysis were missing.

Reviewer #2: Thank you for lending me this opportunity to read this paper. It is insightful and generally well-argued. However, I have several major considerations and suggestions:

1. In my opinion, one of the most fundamental weaknesses in this manuscript is the lack of coherence in the theoretical model. While the authors discussed each path between variables and cite theories to justify individual relationships, the model feels fragmented rather than conceptually unified. It appears as though the mediators (social presence, positive emotions, and empathy) were selected based on convenience rather than a systematic theoretical rationale. This raises a critical concern: Why were these three specific mediators chosen over other potential psychological mechanisms? Currently, the model lacks theoretical integrity such that it feels piecemeal rather than cohesive. The authors cited theoretical justifications for individual pathways, but they did not justify why these mediators should be examined together within the same framework. This suggests that the authors may not have considered how these mediators interact or why they are the most important psychological mechanisms in this context.

As currently presented, positive emotions, social presence, and empathy are conceptually distinct and do not seem to align under a single unifying theme. Therefore, I would suggest the authors to find a way to clearly demonstrate why these variables collectively form the most appropriate psychological mechanisms through which anthropomorphic design influences impulse buying intention. In other words, the authors are suggested to integrate these variables into a coherent framework and explain it clearly in the literature review. For example, the authors may think of categorizing mediators into one or more broader constructs that can align with each other under a single theme. Alternatively, if the authors want to maintain them as distinct variables, they should explain how these three mediators complement each other rather than operating in isolation.

2. In the current version of this paper, moreover, loneliness was treated as a moderator without adequate theoretical grounding. The choice of loneliness as a moderator lacks a compelling justification beyond speculative reasoning. The paper assumes that lonely consumers will respond more strongly to anthropomorphic packaging, but it does not explain why loneliness is the most relevant moderator over other psychological states. Ideally, a good and appropriate moderator should not deviate dramatically from the focal point of the main model/research context. Conceptually, it should be closely linked with IVs and (or) outcome variables. However, I cannot see why loneliness fits in this paper. Moreover, the proposed moderating effect of loneliness (i.e. the effects of anthropomorphic design will be stronger for high-loneliness consumers) is too obvious. Please do not get me wrong here – While intuitive insights can be valuable, readers may question whether this research is actually uncovering new consumer insights or simply confirming common sense. If a moderator’s effect is overly self-evident, it diminishes the contribution of the study because it does not challenge or refine existing theoretical perspectives.

Therefore, if the authors wish to retain loneliness as the moderator, I suggest (1) Explaining clearly why loneliness is the most relevant psychological state/moderator that need to be considered in this research; (2) refining the argument to introduce theoretical complexity in hypotheses development; and (3) Considering testing an additional moderator to strengthen the study’s robustness.

3. The study surveyed 308 beverage consumers in Shenzhen, yet the authors do not provide any justification for why Shenzhen is an appropriate market for this study. This is a critical omission because market characteristics (e.g., consumer demographics, cultural tendencies, purchasing behaviours) can significantly influence the results. Without this justification, it is unclear whether the findings can be generalized to other markets or if Shenzhen represents a unique consumer group.

Additionally, the manuscript did not specify how the sample was recruited. Was it through random sampling, convenience sampling, or quota sampling? The lack of transparency in sampling methodology raises concerns about potential selection bias, which could impact the study’s validity.

The authors also need to justify why a sample size of 308 is appropriate for this study. Given that the proposed model consists of seven constructs, a larger sample would typically be expected to ensure adequate statistical power and model stability in SEM. It would be useful for the authors to conduct and report a power analysis to determine whether 308 responses provide sufficient power to detect meaningful effects.

Lastly, the manuscript misuses the term "experimental results" when describing its findings. Since the study is purely survey-based without an experimental design (i.e. there are no control vs. treatment groups), the term “experimental” is misleading and should be replaced with more appropriate terminology such as “survey findings” or “empirical results” to avoid confusion.

4. While the methodological section is generally solid, the authors failed to test for common method variance, which is a significant concern in survey-based research. To address this, the authors need to include a formal test for CMV, using Marker variable / ULMC / other methods. Please note that Harman’s single-factor test is insufficient and should not be used.

5. The authors did not mention whether control variables were included in the analysis, yet several key consumer characteristics are likely to influence impulse buying behaviours. Specifically, brand familiarity, product experiences, and prior purchasing behaviours are known to have strong effects on consumer decision-making. Without controlling for these factors, the authors risked overestimating the effect of anthropomorphic design because the observed relationships could be partially explained by consumers' pre-existing attitudes / experiences. If control variables were included, the authors need to explicitly state which ones were used, how they were measured, and how they influenced the results. If no control variables were included, the authors must provide a strong justification for why they were omitted and acknowledge this as a limitation.

6. I found that many sections (especially in Theoretical development, Hypotheses, and Results discussion) repeated the same points multiple times. Please streamline the manuscript to remove redundancy. It also seems that the authors put too much emphasis on discussing concepts and theories (e.g., trust, Self-congruity theory, etc.) that irrelevant to the focal points of this research. If I were wrong about this, then please clearly explain how they related to this research. Otherwise, please reduce these discussions which may distract readers. In my opinion, the implications are generic and lack depth. The study concludes that anthropomorphic design can enhance impulse buying, but it fails to translate these findings into concrete marketing strategies that beverage companies can use. Please find a way to strengthen the practical implications, focusing on how beverage companies should specifically implement anthropomorphic design strategies.

I hope you find my suggestions useful. Good luck with your research!

6. PLOS authors have the option to publish the peer review history of their article (what does this mean?). If published, this will include your full peer review and any attached files.

Reviewer #1: No

Reviewer #2: No

---

## [Author Response · Author response to Decision Letter 1]

9 Apr 2025

Dear Reviewers,

Thank you very much for your thorough and constructive feedback on our manuscript titled "Can Anthropomorphic Design in Beverage Packaging Enhance Impulse Buying Intention? Amazing Visual and Verbal Cues!". This study systematically examines how anthropomorphic visual and verbal cues in beverage packaging influence consumers' impulse buying intentions. Grounded in the Stimulus-Organism-Response model, Social Presence Theory, and Empathy Theory, the research employs a consumer survey to validate the mediating roles of positive emotions, social presence, and empathy in the relationship between anthropomorphic design and impulse buying intentions. A comprehensive causal model was developed to explore these relationships. Based on the comments provided by the reviewers, significant modifications have been made to this study, including the establishment of a new model and the relationships between various variables. In addition, more relevant literature was consulted to support this study.

We sincerely appreciate the time and effort you have dedicated to reviewing our work. Your comments have been invaluable in helping us improve the quality and clarity of our paper. Below, we have addressed each of your comments and outlined the revisions we have made in response to your suggestions.

Sincerely,

Yuyang Tian

College of Creative Arts,University Technology Mara (Uitm), Shah Alam, Selangor, Malaysia

---

## [Decision Letter · Decision Letter 1]

8 May 2025

PONE-D-24-59115R1Can Anthropomorphic Design in Beverage Packaging Enhance Impulse Buying Intention? Amazing Visual and Verbal Cues!PLOS ONE

Dear Dr. Tian,

Thank you for submitting your manuscript to PLOS ONE. After careful consideration, we feel that it has merit but does not fully meet PLOS ONE’s publication criteria as it currently stands. Therefore, we invite you to submit a revised version of the manuscript that addresses the points raised during the review process.

Editor's comments:

One of the reviewers still has minor concerns that require your attention. Therefore, I invite you to address what is left on the table. I look forward to receiving your revision.

We look forward to receiving your revised manuscript.

Kind regards,

Simon Dang, Ph.D.

Academic Editor

PLOS ONE

Journal Requirements:

Reviewers' comments:

Reviewer's Responses to Questions

**Comments to the Author**

1. If the authors have adequately addressed your comments raised in a previous round of review and you feel that this manuscript is now acceptable for publication, you may indicate that here to bypass the “Comments to the Author” section, enter your conflict of interest statement in the “Confidential to Editor” section, and submit your "Accept" recommendation.

Reviewer #1: All comments have been addressed

Reviewer #2: (No Response)

2. Is the manuscript technically sound, and do the data support the conclusions?

Reviewer #1: Partly

Reviewer #2: Yes

3. Has the statistical analysis been performed appropriately and rigorously? 

Reviewer #1: N/A

Reviewer #2: Yes

4. Have the authors made all data underlying the findings in their manuscript fully available?

Reviewer #1: Yes

Reviewer #2: Yes

5. Is the manuscript presented in an intelligible fashion and written in standard English?

Reviewer #1: Yes

Reviewer #2: Yes

6. Review Comments to the Author

Reviewer #1: I have no further comments on this version, and I wish you all the best with your publication and ongoing research.

Reviewer #2: The authors have made great efforts to revise the paper as such the quality of the paper has been dramatically improved. I just have one concern:

While the authors stated that they examined the CMV using ULMC method, I did not find the results in the manuscript. While the authors claimed that the results can be found in Lines 647 – 781, I still could not find them. Please follow the following approach to test CMV via ULMC and provide results in a table: Comparing the model fit indices such as CMIN/df, CFI, GFI, TLI, RMSEA, SRMR of the (1) measurement model, (2) measurement model with a ULMC, and (3) model with all items under 1 construct.

Good luck!

7. PLOS authors have the option to publish the peer review history of their article (what does this mean?). If published, this will include your full peer review and any attached files.

Reviewer #1: **Yes: **Trong Huu Nguyen

Reviewer #2: No

---

## [Author Response · Author response to Decision Letter 2]

12 May 2025

Dear Reviewers,

We sincerely appreciate the time and effort you have dedicated to reviewing our manuscript. We are grateful for your constructive feedback, which has significantly improved the quality of our paper. Below, we provide our point-by-point responses to the reviewers’ comments and outline the revisions made to address their concerns. We have carefully addressed all the reviewers’ comments and believe the revised manuscript has been significantly improved. Thank you once again for your time and constructive feedback. We hope the current version meets the journal’s standards for publication. Should you have any further questions or require additional revisions, we would be happy to accommodate them.

Sincerely,

Yuyang Tian

College of Creative Arts, University Technology Mara (Uitm), Malaysia

---

## [Decision Letter · Decision Letter 2]

27 May 2025

Can Anthropomorphic Design in Beverage Packaging Enhance Impulse Buying Intention? Amazing Visual and Verbal Cues!

PONE-D-24-59115R2

Dear Dr. Tian,

We’re pleased to inform you that your manuscript has been judged scientifically suitable for publication and will be formally accepted for publication once it meets all outstanding technical requirements.

Kind regards,

Simon Dang, Ph.D.

Academic Editor

PLOS ONE

Additional Editor Comments (optional):

Reviewers' comments:

Reviewer's Responses to Questions

**Comments to the Author**

1. If the authors have adequately addressed your comments raised in a previous round of review and you feel that this manuscript is now acceptable for publication, you may indicate that here to bypass the “Comments to the Author” section, enter your conflict of interest statement in the “Confidential to Editor” section, and submit your "Accept" recommendation.

Reviewer #2: All comments have been addressed

2. Is the manuscript technically sound, and do the data support the conclusions?

Reviewer #2: Yes

3. Has the statistical analysis been performed appropriately and rigorously? 

Reviewer #2: Yes

4. Have the authors made all data underlying the findings in their manuscript fully available?

Reviewer #2: Yes

5. Is the manuscript presented in an intelligible fashion and written in standard English?

Reviewer #2: Yes

6. Review Comments to the Author

Reviewer #2: All my comments have been addressed. In this case I have no further concerns and recommend publication.

7. PLOS authors have the option to publish the peer review history of their article (what does this mean?). If published, this will include your full peer review and any attached files.

Reviewer #2: No

---

## [Editor Report · Acceptance letter]

PONE-D-24-59115R2

PLOS ONE

Dear Dr. Tian,

I'm pleased to inform you that your manuscript has been deemed suitable for publication in PLOS ONE. Congratulations! Your manuscript is now being handed over to our production team.

Kind regards,

on behalf of

Dr. Simon Dang

Academic Editor

PLOS ONE